# Effective End-Group Modification of Star-Shaped PNVCL from Xanthate to Trithiocarbonate Avoiding Chemical Crosslinking

**DOI:** 10.3390/polym13213677

**Published:** 2021-10-25

**Authors:** Norma A. Cortez-Lemus, Eduardo Hermosillo-Ochoa, Ángel Licea-Claverie

**Affiliations:** Centro de Graduados e Investigación en Química, Tecnológico Nacional de México/Instituto Tecnológico de Tijuana, A. P. 1166., Tijuana 22000, Mexico; eduardo.hermosillo17@tectijuana.edu.mx (E.H.-O.); aliceac@tectijuana.mx (Á.L.-C.)

**Keywords:** aminolysis, star polymers, poly(*N*-vinylcaprolactam), end-group functionality, chain-end functionalization, RAFT, transformation of xanthate to trithiocarbonate

## Abstract

In this study, six-arm star-shaped poly(*N*-vinylcaprolactam) (PNVCL) polymers prepared by reversible addition–fragmentation chain transfer (RAFT) polymerization were subjected to aminolysis reaction using hexylamine. Chemically crosslinked gels or highly end-functionalized star polymers can be obtained depending mainly on the type of solvent used during the transformation of the RAFT functional group. An increase in the viscosity of the solution was observed when the aminolysis was carried out in THF. In contrast, when the reaction was conducted in dichloromethane, chain-end thiol (PNVCL)_6_ star polymers could be obtained. Moreover, when purified (PNVCL-SH)_6_ star polymers are in contact with THF, the gelation occurs in just a few minutes, with an obvious increase in viscosity, to form physical gels that become chemically crosslinked gels after 12 h. Interestingly, when purified (PNVCL-SH)_6_ star polymers were stirred in distilled water, even at high aqueous solution concentration (40 mg/mL), there was no increase in the viscosity or gelation, and no evident gels were observed. The analysis of the hydrodynamic diameter (*D_h_*) by dynamic light scattering (DLS) did not detect quantifiable change even after 4 days of stirring in water. On the other hand, the thiol groups in the (PNVCL-SH)_6_ star polymers were easily transformed into trithiocarbonate groups by addition of CS_2_ followed by benzyl bromide as demonstrated by UV-Vis spectroscopical analysis and GPC. After the modification, the (PNVCL)_6_ star polymers exhibit an intense yellow color typical of the absorption band of trithiocarbonate group at 308 nm. To further demonstrate the highly effective new trithiocarbonate end-functionality, the PNVCL polymers were successfully chain extended with *N*-isopropylacrylamide (NIPAM) to form six-arm star-shaped PNIPAM-*b*-PNVCL block copolymers. Moreover, the terminal thiol end-functionality in the (PNVCL-SH)_6_ star polymers was linked via disulfide bond formation to l-cysteine to further demonstrate its reactivity. Zeta potential analysis shows the pH-responsive behavior of these star polymers due to l-cysteine end-functionalization. By this using methodology and properly selecting the solvent, various environment-sensitive star polymers with different end-groups could be easily accessible.

## 1. Introduction

Polymers prepared by reversible addition–fragmentation chain transfer (RAFT) polymerization can be transformed just by eliminating the RAFT group [1,2,3,4,5]. Although there is a wide variety of possibilities, the removal of the RAFT groups by reaction with nucleophiles, especially the use of primary amines, is the most used method (aminolysis); nevertheless, it is often accompanied by the oxidation of terminal thiol groups resulting in chain–chain coupling, even if the reaction is carried out in the absence of oxygen. The formation of disulfide groups can occur within the removal reaction, in the purification step or during postfunctionalization [3,4,5]. The ability of thiol end-capped polymers to form disulfide bonds strongly depends among many other factors on the R groups attached to it and more specifically on the structure of the monomer [6,7,8,9]. For example, thiol end-groups in poly(*N*,*N*-diethylacrylamide) RAFT-made polymers do not form disulfide groups easily, even though the purification process is long and involves several steps, such as dialysis against THF for 5 days. The SEC chromatogram of the thiolated polymer shows no evidence of macromolecular oxidation, although tris(2-carboxyethyl) phosphine hydrochloride (TCEP) was added after dialysis [7]. In contrast, Winnik et al. [9] reported the aminolysis of poly(*N*-isopropylacrylamide) prepared by RAFT polymerization in the presence of a catalytic amount of TCEP as a reducing agent. The aminolysis was performed in THF at room temperature for 1 h under N_2_ atmosphere. In the GPC traces, a small shoulder on the high molar mass side was observed, which is attributed to disulfide formation, in the presence or absence of TCEP. The authors consider that the oxidation of α,ω-thiol end polymers may occur during the purification of the polymer. The thiol functionality in the polymers was determined in 95% (Ellman test).

After aminolysis of the RAFT-made polymers, several examples prove the successful reactivity of thiol chain-end, e.g., with isocyanates [7], with 2,2,-dithiodipyridine [8] and cholesterol functionalization with alkyl iodides [9].

To date, there is abundant literature related to the aminolysis of linear polymers prepared via RAFT [5,6,7,8,9,10]. However, studies on aminolysis of RAFT-made star-shaped polymers and their postfunctionalization from the thiol are scarce [11,12,13,14,15].

Poly(*N*-vinylcaprolactam) (PNVCL) is a thermosensitive, nonionic polymer that exhibits an LCST in water at around 32 °C. In the case of PNVCL, the transition temperature depends on the molecular weight, polymer concentration and presence of hydrophobic or polar groups at the chain-ends [16,17]. Due mainly to its biocompatibility, this thermosensitive polymer has attracted interest in the biomedical field [18].

In this work, the xanthate end-groups in star-shaped PNVCL polymers were removed using hexylamine (see Figure 1). When the aminolysis reaction was carried out in THF or when the (PNVCL-SH)_6_ star polymers were in contact with THF, chemically crosslinked gels were rapidly formed. In contrast, when the aminolysis reaction was conducted in dichloromethane, (PNVCL-SH)_6_ star polymers with thiol end functionality could be obtained. When these polymers were left to stir in distilled water at ambient temperature for several days, no gel formation was observed. Moreover, the transformation of the reactive thiol chain-end into a trithiocarbonate was also demonstrated. To the best of our knowledge, this transformation on a polymer has not yet been reported. In addition, to further demonstrate the chain-end functionality of the thiols in the (PNVCL-SH)_6_ star polymers, l-cysteine was coupled via disulfide bond formation.

## 2. Experimental

### 2.1. Materials

2-Bromopropionyl bromide (Aldrich, 97%), dipentaerythritol, (DPERT, 85%, Aldrich, St. Louis, MO, USA), triethylamine (TEA, 99.5%), 4,4-azobis(4-cyanovaleric acid) (ACVA) (Fluka, St. Louis, MO, USA), 2,2′-azobis(4-methoxy-2,4-dimethyl valeronitrile) (V-70, Wako, 96%, Chemicals, Richmond, VA, USA, 96%), carbon disulfide (99.9%, Sigma Aldrich), anhydrous sodium sulphate (Na_2_SO_4_), potassium ethyl xanthogenate (Aldrich, 97%), *N*-isopropylacrylamide (97%, Sigma Aldrich), benzyl bromide (Sigma-Aldrich, 98%), hexylamine (Aldrich, 99%), L-cysteine (Sigma-Aldrich, 97%) and NVCL (Aldrich, >99%) were purified by recrystallization with diethyl ether. Hydrochloric acid (HCl, 10% aqueous solution, Fermont, Monterrey, Mexico), ethanol (Fermont, 99%), tetrahydrofuran (THF, Fermont, 99.9%), dimethylformamide (DMF, Fermont, Monterrey, Mexico, 99.9%), chloroform (Sigma-Aldrich, 99.5%) and *p*-dioxane (Sigma-Aldrich, 99.8%) were used as received. Column chromatographic purifications were performed using silica gel (70–230 mesh, Acros Organics, Branchburg, NJ, USA).

### 2.2. Measurements

Dynamic light scattering (DLS) measurements were carried out using a Malvern Instruments Nano-ZS Nanosizer (ZEN 3690). The instrument is equipped with a helium–neon laser (633 nm) with a size detection range of 0.6 nm–5 μm. DLS experiments were performed at the scattering angle of 90° and equilibrated for 10 min before data collection. The solutions/dispersions were filtered through a 0.45 μm nylon membrane filter before analysis to remove dust. The volume-average hydrodynamic diameter (*D_h_*) and polydispersity index (PDI) were calculated using Malvern Instruments dispersion technology software, based on CONTIN analysis and Stokes–Einstein equation for spheres as usual. The LCST was taken as the temperature at which the star polymer was still soluble (just before the solution started to turn cloudy) at a 1 mg/mL concentration in water. The LCST was measured by DLS using the Nano-ZS Nanosizer with a temperature program that increased from 20 to 50 °C in two-degree steps, equilibrating for 4 min once the measurement temperature was achieved; measurements were performed three times, each of which included three 30 s runs. Gel permeation chromatography (GPC) was performed on a Varian 9002 chromatograph equipped with a series of three columns (Phenogel: OH-646-K0, OH-645-K0 and OH-643-K0) and two detectors: a refractive index detector (Varian RI-4) and a triangle light scattering detector (LS detector MINI-DAWN, Wyatt). The measurements were performed in THF at 35 °C. Polystyrene standards were used for calibration of the LS detector. THF was used as the mobile phase at a flow rate of 0.7 mL/min. Sample solutions were prepared using 20 mg/mL concentration and filtered through a 0.45 μm PTFE membrane filter before analysis.

^1^H and ^13^C NMR spectra were collected on a Bruker AMX-400 (400 MHz) spectrometer and are reported in ppm using TMS as internal standard. The solvent used was deuterated chloroform (CDCl_3_) for all samples. UV-Vis spectra of the PNVCL polymers and copolymers were recorded using a UV-Vis Varian Cary 100 Spectrophotometer at room temperature.

### 2.3. Synthetic Methods

#### 2.3.1. Preparation of Linear and Star PNVCL Polymers

The synthesis of the linear and star PNVCL polymers was performed following the methodology reported previously [17,19]; see Appendix A.

#### 2.3.2. Aminolysis of PNVCL Polymers Containing Xanthate End-Groups

In a 20 mL vial containing a magnetic stir bar, 0.5 g of (PNVCL_48_)_6_ star polymers (0.5 g, 0.0126 mmol, *M_n_* _GPC_ = 39,610 g/mol and *Ð* = 1.15, see Figure 1b, Appendix A, entry 1) was dissolved in 6 mL of CH_2_Cl_2_ and hexylamine (0.08 mL) was added. The reaction mixture was stirred at room temperature for 10 min, then the polymer was precipitated by adding an excess of ethyl ether/petroleum ether (1:1), maintaining vigorous agitation. During the entire purification process, the vial must remain capped. The liquid phase was decanted, and the polymer product was redissolved in the minimum amount of dichloromethane and precipitated again with ethyl ether/petroleum ether (1:1); this procedure was repeated three times. In this step, the excess of hexylamine and the by-products were removed. The polymer product was dried under vacuum (always at room temperature). These purification processes must be carried out within 15 min, and it is recommended to start the next transformation step as soon as possible, although the thiol-terminated polymer can be stored in a freezer without any change for up to two weeks. To analyze the sample on the GPC equipment, one purification is sufficient. The (PNVCL_48_-SH)_6_ star polymers were obtained as white solids (80%), *M_n_*
_GPC_ = 47,260 g/mol and *Ð* = 1.07 (see Figure 1b).

#### 2.3.3. Recuperation of O-Ethyl Hexylcarbamothioate

The solvent waste from the first reprecipitation of the aminolysis reaction carried out with petroleum ether/ethyl ether was concentrated in vacuo. The product was purified using column chromatography on silica gel (CH_2_Cl_2_:hexanes 3:1). A thick yellowish oil was obtained.

#### 2.3.4. Transformation of (PNVCL-SH)_6_ Star Polymers into (PNVCL-Trithiocarbonate)_6_ Star Polymers

The sample (PNVCL_32_)_6_ was the precursor for this reaction (*M_n_*
_GPC_ = 26,900 g/mol, *Ð* = 1.13, Appendix A, entry 2). In a 20 mL vial containing a magnetic stir bar, (PNVCL-SH)_6_ star polymers (0.5 g, 0.019 mmol) were dissolved in 5 mL of CHCl_3_. Then, triethylamine (0.031 mL, 0.21 mmol) was added, followed by the addition of CS_2_ (0.018 mL, 0.3 mmol). After stirring for 3 min, benzyl bromide (0.025 mL, 0.21 mmol) was added. The reaction was stirred for an additional 2 h at room temperature. The reaction product was precipitated by adding an excess of ethyl ether, maintaining vigorous agitation. The solvent was discarded, and the resulting polymer was redissolved in the minimum amount of dichloromethane and precipitated again with ethyl ether; this procedure was repeated three times. The polymer product was dried in vacuo. Then, the polymer was redissolved in THF, and the salt was removed by ultracentrifugation cycle (12,000 rpm for 4 min). This purification method is more efficient than washing with dilute HCl and NaHCO_3_. The star polymer containing trithiocarbonate-type functional groups was obtained as a yellow solid (80%), (*M_n_*
_GPC_ = 27,500 g/mol, *Ð* = 1.2).

#### 2.3.5. Chain Extension Polymerization of (PNVCL)_6_ Trithiocarbonate-Type Star Polymers with NIPAM

The starting xanthate polymer for this reaction was (PNVCL_55_)_6_ (*M_n_*
_GPC_ = 46,100 g/mol, *Ð* = 1.11, see Appendix A, entry 3). Six-arm star-shaped (PNVCL_55_)_6_ trithiocarbonate-type polymers (macro-RAFT agent) (*M_n_*
_GPC_ = 46,120 g/mol, *Ð* = 1.12) (148 mg, 0.00325 mmol), NIPAM (74 mg, 0.65 mmol) and ACVA (0.2 mg in 1 mL of DMF, 6.5 × 10^−4^ mmol) were dissolved in 3 mL of DMF. The NIPAM/macro-RAFT agent/ACVA ratio was 200/5/1. The solution was deoxygenated by bubbling nitrogen for 20 min at room temperature. Then, the flask was placed in an oil bath preheated at 70 °C. After 12 h, the polymerization was stopped by cooling to room temperature. Then, the polymer was precipitated by adding an excess of ethyl ether, maintaining vigorous agitation. The resulting polymer was dried under vacuum and redissolved in the minimum amount of dichloromethane and precipitated again with ethyl ether; this procedure was repeated three times. Finally, the copolymer product was dried under vacuum. The polymer, star (PNVCL_55_-*b*-PNIPAM_19%_)_6_ (*M_n_*
_GPC_ = 47,320 g/mol, *Ð* = 1.27), was obtained as a yellowish solid (35%).

#### 2.3.6. Functionalization of PNVCL-SH Polymers with l-Cysteine through the Formation of Disulfide Groups

In a 20 mL vial containing a magnetic stir bar, 0.5 g of star (PNVCL-SH)_6_ polymers (*M_n_* _GPC_ = 29,200 g/mol, *Ð* = 1.24, data corresponding to the precursor xanthate polymer, Appendix A, entry 4) were first dissolved in 1 mL of CH_2_Cl_2_, followed by the addition of l-cysteine (0.1 g, 0.82 mmol). After 5 min, 3 mL of ethanol was added. The reaction mixture was stirred for 72 h. The remaining l-cysteine was removed by ultracentrifugation (12,000 rpm for 4 min). The solvent was removed under reduced pressure. The polymer was redissolved in dichloromethane, and an ultracentrifugation cycle was carried out to remove traces of l-cysteine. This last step is important to gain the total solubility of the polymer product in THF before its analysis in GPC. The resulting polymer was dried under vacuum. (PNVCL-S-S-cysteine)_6_ star polymers (*M_n_*
_GPC_ = 43,560 g/mol, *Ð* = 1.22) were obtained with 95% yield.

## 3. Results and Discussion

### 3.1. Aminolysis of Linear and Star PNVCL Polymers Containing Xanthate End-Groups and Their Transformation into Polymers Containing Thiol End-Groups

The generation of reactive terminal thiol polymers through an aminolysis reaction with the aim of postfunctionalization is quite relevant for precision polymer synthesis. To achieve this, it is necessary to avoid the formation of disulfide bonds between the polymer chain ends. Some of the influencing factors could be the chemical structure of the polymer, the amine type, the solvent, the reaction time and the atmosphere (open air, N_2_) [1,2,3,4,5].

According to a review of the literature, the removal of the thiocarbonylthio group is mostly carried out in THF as solvent. Therefore, based on these facts, this study began seeking the aminolysis of the xanthate group in the RAFT-made (PNVCL)_6_ star polymers. The xanthate group was removed using hexylamine (8-fold in excess for each arm) in THF solution, at ambient conditions. It is important to mention that the concentration of the polymer in the solution was 37 mg/mL.

An aliquot of polymer solution was taken out at different times, purified (as described in Section 2.3.2, experimental part) and analyzed by GPC. In Figure 1, the GPC chromatograms for (PNVCL-SH)_6_ star product and the starting (PNVCL-xanthate)_6_ star polymers are shown. The thiol-containing samples taken out at 5, 10 and 15 min of reaction show a prominent shoulder on the high-molar-mass side of the chromatogram which can be attributed to the formation of disulfide groups between polymer chains. Moreover, the intensity of the main peak decreases as the reaction time increases. It is important to note that the main peak for the (PNVCL-SH)_6_ star polymers is shifted to larger retention times (see sample at 5 min of reaction, Figure 1a); more details will be discussed later. Within 40 min of reaction, visual inspection indicated an evident increase in the viscosity which made it impossible to continue the analysis by GPC. After 4 h, gelation was evident in the vial.

Probably, thiol end polymers are most prone to side reactions in the presence of THF because this solvent promotes oxidation. For instance, it was observed that RAFT agents show a loss of color in the presence of THF associated with the tendency of this solvent to form peroxides. It is reported that the color change was a consequence of the oxidation of the thiocarbonylthio group (S=C–S–) into a sulfine group (O=C–S–) [5,6].

Although the removal of the RAFT groups through aminolysis in a DMF solution has been reported in the literature, it is not recommended here, because the polymer purification process after aminolysis is complicated due to the inherent polarity of DMF, delaying the polymer purification. Dichloromethane was the selected solvent for the aminolysis reaction of the polymers. The reaction time was set at only 10 min because at 5 min there is no longer an absorbance of the thiocarbonylthio group related to the xanthate according to the UV-Vis analysis, as will be shown later.

Based on the literature, the reaction for the removal of the RAFT group by aminolysis can be carried out in times ranging from seconds to several hours [20,21,22,23]. RAFT-made polymers containing a xanthate group are very susceptible to hydrolysis [20] or oxidation [21] compared to those with trithiocarbonate [22], especially in the case of methacrylate-type polymers [23].

The GPC analysis procedure for thiol-functionalized star polymers also had to be optimized since as soon as the samples are exposed to THF (*c* = 20 mg/mL), the formation of disulfide groups seems to start rapidly. After purification, the sample was dissolved in THF and stirred only for 30 s; then, it was passed through the filter and quickly injected into the GPC equipment.

As mentioned previously, the THF GPC traces of the (PNVCL-SH)_6_ star polymers generated in dichloromethane (see Figure 1b) were always shifted to longer retention times compared with the main peak of the starting polymer, probably due to the strong interaction of the thiol fragment with the column [24]. As it can be seen, the GPC curve was unimodal, symmetrical and not skewed, although wider as compared to the starting polymer. As will be discussed and demonstrated later, after postfunctionalization, the GPC curve of functionalized polymers has practically the same retention time as that of the starting polymer. It is important to note the “inconsistency” with the molecular weights calculated by GPC. The (PNVCL-SH)_6_ star polymers show a higher “apparent molecular weight” than the starting polymer. This trend was observed for all the thiol-containing samples analyzed in GPC in this work. For example, the *M_n_*
_GPC_ for the starting polymer and the corresponding terminal thiol polymer was 39,610 and 47,260 g/mol, respectively (see Figure 1b). On the one hand, purified (PNVCL-SH)_6_ star polymers showed good stability when left stirring in dichloromethane for 48 h. The ^1^H NMR spectra in Appendix A display the “freshly” obtained sample and the sample after 48 h of stirring. No additional signals to the typical spectrum of the PNVCL are observed.

On the other hand, the efficient removal of the xanthate group was also proved with the recuperation of the carbamodithiate, which is an important by-product of the aminolysis reaction. The hexylamine attacks the carbonyl thio of the xanthate group, producing two rotamers of *O*-ethyl hexylcarbamothioate (See Figure 2). The ^1^H NMR spectrum (See Figure 2) reveals two quartets at 4.48 and 4.40 ppm, “e”, corresponding to the methylene group attached to the oxygen. The hydrogen from the amine “c” appears at 6.53 and 6.12 ppm. Two quartets corresponding to the methylene attached to the amine “d” at 3.46 and 3.18 ppm are also observed. These compounds have not been previously reported as by-products of the aminolysis reaction. This type of compound has been obtained in cress extracts [25]. Thiocarbamates are also described as additives in insect repulsive agents [26].

In the mass spectrum, the peak of the molecular ion of 189 *m*/*z* can be clearly observed, which is also the base peak, corresponding to the molecular weight of the compound. This further demonstrates the structure of the *O*-ethyl hexylcarbamothioate (See Appendix A).

On the other hand, for comparison purposes, aminolysis was also carried out on linear PNVCL polymers using the same optimized conditions reported above (using dichloromethane as solvent). Solutions of linear polymers in THF were prepared and left stirring for different periods of time (1 min and 5, 10 and 24 h) to later be analyzed by GPC. In Figure 3, it can be observed that the chromatogram position of the PNVCL-SH sample with 1 min of stirring in THF also moves to longer retention times, although the displacement is less marked compared with the starting polymer (again, due to interaction of the thiol groups with the column). It should be noted that for the sample at 1 min stirring, a higher molecular weight than the starting polymer (PNVCL-xanthate *M_n_*
_GPC_ = 14,400 g/mol vs. PNVCL-SH *M_n_*
_GPC_ = 15,600 g/mol) was observed (see Figure 3). This inconsistency has been observed during the GPC analysis of all the terminal thiol end-capped PNVCL polymers, regardless of the topology.

The sample analyzed at 5 h was bimodal with a shoulder at lower retention time. The associate peak has almost the same intensity that the main peak. For the sample at 10 and 24 h, it is observed that the intensity of the main peak decreases while the associated peak increases. Note that the chromatogram for the sample at 24 h (*M_n_*
_GPC_ = 31,000 g/mol, *Ð* = 1.35) exhibits practically twice the molecular weight with respect to the starting PNVCL polymer.

### 3.2. D_h_ Behavior of Linear PNVCL-SH and Six-Arm (PNVCL-SH)_6_ Star Polymers in THF and Water

After the aminolysis reaction, samples of six-arm (PNVCL-SH)_6_ star and linear PNVCL-SH polymers were dissolved in distilled water or THF at varying polymer concentrations and analyzed at different times to study the evolution of *D_h_* by DLS at 25 °C. Figure 4a shows the *D_h_* of the (PNVCL-SH)_6_ star polymers after stirring 10 min in THF. Regardless of polymer concentration, the *D_h_* was around 8–10 nm. The samples were kept stirring in THF, and the *D_h_* was analyzed after 2 and 72 h. In the case of the star-shaped (PNVCL-SH)_6_ polymers, the sample at 1 mg/mL exhibits *D_h_* values of 32 nm after 2 h and 59 nm after 72 h of stirring (Figure 4a). The (PNVCL-SH)_6_ sample at concentrations of 5, 10 and 20 mg/mL was not analyzed after 2 h of stirring because a marked increase in the viscosity of the solution was observed after 6 h, and a gel was formed in the case of the sample at concentrations of 10 and 20 mg/mL. In the case of the sample at concentrations of 1 and 5 mg/mL, no gel formation was observed, even after 5 days. However, the increase in viscosity in the solution was obvious by visual inspection. These results are consistent with what was observed during the aminolysis reaction in THF for the (PNVCL-SH)_6_ star polymers (see Appendix A, entry 1), thereby confirming the strong effect of the solvent selection on the formation of disulfide bonds. In contrast, the behavior of the six-arm (PNVCL-SH)_6_ star polymers dissolved in water was totally different, as confirmed by following the evolution of *D_h_*. The samples were prepared with varying concentrations, and the measurements were carried out after 1 and 72 h of stirring. It was observed that the *D_h_* remained practically unchanged (8 to 10 nm, Figure 4b) in all polymer samples. For clarity, only data taken at 72 h are shown. The encountered behavior is summarized in Figure 3.

On the other hand, the behavior of linear PNVCL-SH polymers in THF solution is very different from that of star-type polymers. The change in the hydrodynamic diameter is practically negligible even when varying the concentrations of polymer in THF solution (Figure 5a). With respect to polymers dissolved in water at different concentrations, the behavior is like that of star-type polymers. The *D_h_* remained practically unchanged (Figure 5b) in all polymer samples. For clarity, only data taken at 72 h are shown.

### 3.3. Easy and Fast Transformation of (PNVCL-Xanthate)_6_ Stars into (PNVCL-Trithiocarbonate)_6_ Star Polymers and the Synthesis of (PNVCL-b-PNIPAM)_6_ Block Copolymers

The xanthate group on star PNVCL polymers was removed by aminolysis using hexylamine. Addition of triethylamine to (PNVCL-SH)_6_ star polymers followed by CS_2_ generates the trithiocarbonate anion (see Figure 4). The polymer solution immediately attained an intense yellow color, and then the selected bromide (in this case benzyl bromide) was added. Attempts to use 2-bromopropionate instead to generate a good living group were not successful. The yellow color was lost in the solution during the purification process, and the precipitated polymer was practically colorless. When this polymer was analyzed by UV-Vis spectroscopy, the typical absorption band at 308–309 nm for trithiocarbonates was very weak, which was indicative of a low degree of functionalization. A possible explanation for this behavior is that the reaction is reversible, returning to the original reagents, which has been reported by some research groups [12].

In contrast, benzyl bromide reacts with the trithiocarbonate anion very efficiently, resulting in a yellow polymer after the purification step. The xanthate moiety and the trithiocarbonate moiety have different UV absorptions from each other. In Figure 6a, UV-Vis spectra show the absorbance band at 284 nm for the PNVCL xanthate-type polymer. After the aminolysis, this band disappears completely in the thiol end-capped polymer. The typical absorbance at 308 nm is observed for the PNVCL polymers end-functionalized with the trithiocarbonate group.

In Figure 6b, the GPC traces of (PNVCL)_6_ star polymers with xanthate, thiol and trithiocarbonate groups are shown. The chromatogram of the thiol end sample (*M_n_*
_GPC_ = 31,190 g/mol, *Ð* = 1.25) is shifted at a longer retention time and slightly higher molecular weight, but after the postfunctionalization, the position of the peak of the trithiocarbonate polymer (*M_n_*
_GPC_ = 27,500 g/mol, *Ð* = 1.2) was practically the same as that of the starting xanthate polymer (*M_n_*
_GPC_ = 26,900 g/mol, *Ð* = 1.13, Appendix A, entry 2), although the dispersity increased slightly. The possibility of unwanted thiol coupling is always present, especially during the purification process of the thiol end-capped polymers.

The ^1^H NMR spectrum for the (PNVCL-trithiocarbonate)_6_ star polymer is shown in Figure 7. At 7.33 ppm a multiplet peak is observed that corresponds to aromatic hydrogens, and its presence proves the efficient functionalization of the trithiocarbonate group in the polymer.

### 3.4. Chain Extension Polymerization of the (PNVCL-Trithiocarbonate)_6_ Star Polymers with NIPAM

The preparation of block copolymers involving “less activated” monomers (LAM) and “more activated” monomers (MAM) by controlled radical polymerization is still a challenge [27,28,29]. To date, in the literature, a chain extension of PNVCL with MAMs is scarcely found. Recently, Zhang and Cheng et al. [29] reported the synthesis of poly(*N*-vinylpyrrolidone)-based block copolymers by RAFT polymerization under irradiation of UV or visible light. The PMAMs (butyl acrylate, methyl acrylate and styrene) were used as the macro-RAFT agents. In addition, the living character of polymerization was demonstrated.

The living character of the (PNVCL)_6_ star polymers transformed from xanthate into trithiocarbonate-functional groups was demonstrated with the polymerization of *N*-isopropylacrylamide. The polymerization was performed in DMF at 70 °C using ACVA as an initiator.

The ^1^H NMR spectrum displayed in Figure 8 shows the characteristic peaks corresponding to PNVCL and PNIPAM: the methine peak (–NC*H*–) of the PNVCL block is observed at 4.39 ppm “*b*”. For the PNIPAM block, the hydrogen of the amide group (–N*H*–) “*d*” was observed at 7.04–6.0 ppm, and the methine (–NHC*H*–) “*c*” was observed at 4.0 ppm. The molar composition of each block was determined by comparing the integration value of the methine peak “*b*” of the PNVCL block at 4.49 ppm and the methine peak for PNIPAM at 4.0 ppm “*c*”. This calculation demonstrated that the copolymer has a PNIPAM content of 19%.

With this simple and fast transformation of xanthates to trithiocarbonates presented here, the possibility of generating a great variety of block copolymers is opened by combining LAM monomers such as NVCL and MAM monomers such as methacrylates.

In Figure 9, the GPC traces for the (PNVCL-trithiocarbonate)_6_ star polymers (*M_n_*
_GPC_ = 46,120 g/mol, *Ð* = 1.12) and the (PNVCL-*b*-PNIPAM)_6_ star block copolymers (*M_n_*
_GPC_ = 47,320 g/mol, *Ð* = 1.27) are shown. As can be seen, the GPC chromatogram for the star block copolymer was shifted to shorter retention times compared to the starting star polymer.

The transition temperature behavior for (PNVCL_55_)_6_ star polymers and (PNVCL_55_-*b*-PNIPAM_19%_)_6_ star block copolymers was evaluated for both heating and cooling cycles. Figure 10a shows the hysteresis of about 2 °C for (PNVCL_55_)_6_ star polymers. PNVCL does not show hysteresis normally [30,31], unlike PNIPAM [32]. As has been discussed in the literature, the PNVCL transition temperature is very sensitive to the end groups, and this sensitivity is even greater with a six-arm star polymer. It should be noted that the transition temperature of the star block copolymer was 32 °C (Figure 10b), while that of the starting PNVCL star polymer was 36 °C. This change can be attributed to the PNIPAM content (19%) in the copolymer, and the hysteresis is around 2 °C (almost negligible). In contrast, the hysteresis observed for the (PNVCL_55_-*b*-PNIPAM_50%_)_6_ star block copolymer is remarkable for the cooling process (Figure 9); as the PNIPAM content increases to 50%, it gives the copolymer a hysteresis of 10 °C.

### 3.5. Functionalization of PNVCL-SH Polymers with l-Cysteine through the Formation of Disulfide Groups

The thiol end-groups in the star-shaped (PNVCL-SH)_6_ polymers were reacted with l-cysteine through the formation of disulfide groups (4 equiv excess by arm) by stirring in ethanol or DMF for 72 h.

The sample of (PNVCL-S-S-cysteine)_6_ functionalized star polymer was dissolved in THF (20 mg/ mL) and was left under observation for 24 h to be analyzed later by GPC. We mentioned above that the thiol-terminated polymers dissolved in THF tend to form gels quickly. So, the fact that the sample could be analyzed without problems by GPC proves the transformation of the thiol group.

In Figure 11a, the chromatograms of the polymer functionalized with L-cysteine and the starting polymer are shown. As can be seen, the GPC curve for the star PNVCL-S-S-Cys sample is unimodal and is shifted to longer retention times with respect to the starting polymer. Once functionalized, the end-chain polymer has amino and carboxyl groups (by arm) that contribute to its retention in the column. In addition, the molecular weight of the (PNVCL-S-S-cysteine)_6_ star polymer was apparently increased (*M_n_* _GPC_ = 43,560 g/mol, *Ð* = 1.22) with respect to the starting xanthate polymer (*M_n_*
_GPC_ = 29,200 g/mol, *Ð* = 1.24, Appendix A, entry 4).

Other evidence of functionalization with l-cysteine is the modification of the transition temperature of the PNVCL polymers. The great sensitivity of LCST in the PNVCL due to chain-end groups/molecular weight has been mentioned previously. This behavior is more marked for low-molecular-weight star-shaped PNVCL polymers [17]. The selected thiol-terminated polymer for this study, (PNVCL-SH)_6_, has an LCST of 30 °C. It can be observed in Figure 11b that the LCST increased to 34 °C after end-group functionalization with cysteine. The pH-responsive behavior (PNVCL-S-S-cysteine)_6_ sample was studied by measuring the zeta potential value using DLS. The (PNVCL_35_-xanthate)_6_ and the thiol end polymer were also studied (Figure 11c). The star (PNVCL-S-S-cysteine)_6_ in aqueous solution at pH 2.25 showed a zeta potential value of +3 mV (because of the protonation of the amine groups in the chain end). As the pH increased, the value of the zeta potential became negative. For example, at a pH value of 8.57, the zeta potential value was strongly negative: −29.9 mV. The pH-responsive behavior of the (PNVCL-S-S-cysteine)_6_ was higher compared with the thiolated and the (PNVCL_35_-xanthate)_6_ star polymer. In the thiol end sample (dissolved in distilled water, 2.5 mg/mL), the zeta potential value was −9.81 mV (at pH value of 7.24, unadjusted (see Appendix A)), and it changed very slightly at higher pH values (Figure 11c). Moreover, although the PNVCL is considered a nonionic polymer, in this study it was observed that when varying the pH value, there is a change in the zeta potential, although the response is much lower compared to the cysteine-functionalized polymer. The (PNVCL_35_-xanthate)_6_ star polymer dissolved in distilled water presented a pH value of 7.01 and a zeta potential value of −10.2 mV (see Appendix A). PNVCL prepared via free radicals also presented a zeta potential value (−10.8 mV) like that of the star-type polymer prepared via RAFT (see Appendix A). In any case, at the pH of 8.57, the xanthate-functionalized star showed a zeta potential of −12.5 mV, in contrast with the −29 mV value for the cysteine-functionalized star polymer. The change in zeta potential with a pH change is clearly enhanced by the presence of the cysteine end-groups in the star polymer.

## 4. Conclusions

Thiol-terminated star-shaped poly(*N*-vinylcaprolactam) polymers were prepared by aminolysis reaction using hexylamine in dichloromethane as solvent. (PNVCL-SH)_6_ polymers are precursors of either cross-linked gels or highly effective postfunctionalized star polymers, depending on the selected solvent. When the thiol end-capped star polymers are in contact with THF, they form gels rapidly. In contrast, (PNVCL-SH)_6_ polymers dissolved and stirred for days in distilled water do not form gels. The thiol groups in the (PNVCL-SH)_6_ star polymers were transformed into trithiocarbonate RAFT groups. After the modification, the star polymers exhibit an intense yellow color typical of the absorption band of the trithiocarbonate group at 308 nm. The (PNVCL-trithiocarbonate)_6_ star polymers were successfully chain extended with *N*-isopropylacrylamide (NIPAM) to form six-arm star-shaped PNIPAM-*b*-PNVCL block copolymers. Moreover, the terminal thiol end-functionality in the (PNVCL-SH)_6_ star polymers was linked via disulfide bond formation to L-cysteine to further demonstrate its reactivity. Zeta potential analysis shows the pH-responsive behavior of these star polymers due to the L-cysteine end-functionalization. The LCST values of the PNVCL star polymers change as expected with molecular weight, end-group functionalization and comonomer.

## Data Availability

Not applicable.

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
