# Peer review of "Effective End-Group Modification of Star-Shaped PNVCL from Xanthate to Trithiocarbonate Avoiding Chemical Crosslinking"

_polymers, 2021, doi:10.3390/polym13213677_

Round 1
Reviewer 1 Report
The authors examine an interesting question in the area of RAFT polymers: what happens to star-shaped polymers upon aminolysis of the chain-end thioester functionality. They find that it is dependent upon the solvent. In THF they obtain evidence for cross linking and in water no gelation is observed. I think the work is of high relevance to polymer chemistry and results are interesting for the field. The technical quality of the manuscript is good and the figures and the description is clear. Therefore, I recommend acceptance as is.
Author Response
Thank you very much for your review and for the good comments on this work.
Reviewer 2 Report
This manuscript reports the amimolysis effect on star shaped poly(N-vinylcaprolactam polymers prepared via RAFT polymerization. The experimental parts is clearly written as the entire paper. The GPC curves must be absolutely normalized in order to make a direct comparison of polymer's polydispersity. After this correction, the paper will be ready for publication in Polymers.
Author Response
Thank you very much for your review and for the good comments on this work. According to your suggestion, all the chromatograms in this revised version were normalized, to gain more clarity.
Reviewer 3 Report
The manuscript submitted to Polymers entitled “ What to Expect after Aminolysis of Star-Shaped Poly(N-Vi-nylcaprolactam) Polymers Prepared by RAFT: Efficient End-Functionalization with the Transformation of Xanthate Groups or Chemical Crosslinking?” aminolysis reaction of six arms star-shaped poly(N-vinylcaprolactam) (PNVCL) polymers synthesized by RAFT Polymerizatin method. Authors demodstrated the selectivity of solvent used during functionalization for formation of chemically crosslinked gels or functionalized six arm crosslinked polymers. In principle, manuscript is well written and logically organized. I would recommend the manuscript for publication after some modifications.
- Title is unnecessarily lengthy and even confusing. Make it concise and comprehensive.
- I note flaws in language that makes it difficult for readers at some points, need to revise
- Introduction should be re-written to properly build the case and narrative for justification of this study.
- Should elaborate more why elution volume increase initially of the parent polymer and then decrease after giving more reaction time- Figure 1A, Figure 3, and why behavior is opposite for polymer shown in Figure 1B
- Is it a fair comparison in Figure 6?
Author Response
We thank very much to the Reviewer 3 for the suggestions, contributions and review of this manuscript.
- Title is unnecessarily lengthy and even confusing. Make it concise and comprehensive.
Authors:
The title has been modified.
The new title of this work is:
Effective end-group modification of star shaped PNVCL from xanthate to trithiocarbonate avoiding chemical crosslinking
- I note flaws in language that makes it difficult for readers at some points, need to revise.Authors. The manuscript was revised, finding few details that are marked in red.
- Introduction should be re-written to properly build the case and narrative for justification of this study.
Authors:
A paragraph was deleted from the introductory part. And a whole fragment was transferred from the introduction part to the section 3.4 of the manuscript, to gain clarity.The references also had to be re-organized.
4. Should elaborate more why elution volume increase initially of the parent polymer and then decrease after giving more reaction time- Figure 1A, Figure 3, and why behavior is opposite for polymer shown in Figure and 5 Is it a fair comparison in Figure 6?
Authors:
As described in the discussion part, all thiol end polymer samples were dissolved in THF and stirred only for 30 seconds, and quickly injected into the GPC equipment. Sample preparation is done as quickly as possible. If there is a significant delay, shoulders are immediately seen at high molecular weights.
With respect to the curve of thiol end sample in the Figure 6. There are samples that are less retained, but this is more related to the preparation and handling of the sample for GPC analysis.